# Determination of Fumonisin B_1_ by Aptamer-Based Fluorescence Resonance Energy Transfer

**DOI:** 10.3390/s22228598

**Published:** 2022-11-08

**Authors:** Xinyue Zhao, Jiale Gao, Yuzhu Song, Jinyang Zhang, Qinqin Han

**Affiliations:** Faculty of Life Science and Technology, Kunming University of Science and Technology, Kunming 650500, China

**Keywords:** fumonisin B_1_, aptamer, fluorescence resonance energy transfer, graphene oxide, corn flour

## Abstract

Fumonisin FB is produced by *Fusarium moniliforme* Sheld, of which FB_1_ is the most common and the most toxic. The establishment of a rapid detection method is an important means to prevent and control FB_1_ pollution. A highly sensitive fluorescent sensor based on an aptamer for the rapid detection of fumonisin B_1_ (FB_1_) in corn was established. In this study, 5-carboxyfluorescein (FAM) was labeled on the aptamer of FB_1_ (F10). F10 was adsorbed on the surface of graphene oxide (GO) by π-π stacking. The FAM fluorescence signal could be quenched by fluorescence resonance energy transfer between fluorescent molecules and graphene oxide (GO). In the presence of FB_1_, the binding efficiency of the aptamer to GO was reduced. Therefore, the content of FB_1_ in corn samples was determined by fluorescence measurements of mixed FAM-labeled F10, GO and corn samples. This method had a good linear relationship in an FB_1_ concentration range of 0–3000 ng/mL. The equation was y = 0.2576x + 10.98, R^2^ = 0.9936. The limit of detection was 14.42 ng/mL, and the limit of quantification was 43.70 ng/mL. The recovery of a spiked standard in the corn sample was 89.13–102.08%, and the time of detection was 30 min.

## 1. Introduction

Fumonisin FB, produced by *Fusarium moniliforme* Sheld, is an important mycotoxin. Several mycotoxin types exist, namely fumonisin A_1_ (FA_1_), fumonisin A_2_ (FA_2_), fumonisin B_1_ (FB_1_), fumonisin B_2_ (FB_2_), fumonisin B_3_ (FB_3_), and others, of which FB_1_ is the most common and the most toxic [1]. It has been classified as a class 2B carcinogen by the International Agency for Research on Cancer (IARC) [2,3]. China’s National Food Safety guidelines do not specify a limit for FB_1_ in food, but the European Union and the U.S. Food and Drug Administration have set the upper limit for FB_1_ in feed as 200–2000 μg/kg and 3000–4000 μg/kg, respectively [4]. Corn is the crop most polluted by FB_1_, which has also been detected in wheat and soybeans. As corn is the most common crop in animal feed, pollution can lead to serious adverse events in animals eating contaminated feed [2]. The mechanism of FB_1_ toxicity is based on the inhibition of sphingolipid biosynthesis, which is an important cellular event, and the inhibition of sphingolipid biosynthesis can lead to organ failure [5,6]. Existing studies have demonstrated that a variety of diseases in mammals can be caused by FB_1_, such as equine white matter encephalomalacia [7], porcine pulmonary edema [8], mouse liver tumor development [9], and acute kidney poisoning in goats [10]. FB_1_ can also cause serious adverse events in chickens and other types of poultry, such as sudden weight loss, shrinkage of organs (e.g., spleen), acute myocarditis, liver injury, and increased mortality [11,12,13]. Therefore, the establishment of a method for the detection of possible FB_1_ residues in food samples is of great importance for food safety monitoring.

The enzyme-linked immunosorbent assay (ELISA) is a standard analytical technique used to monitor mycotoxin pollution in food and animal feed. Shu et al. established a one-step competitive ELISA to quickly and efficiently detect FB_1_ in grain samples. The limit of detection (LOD) of the colorimetric ELISA was 0.35 ng/mL, and there was a good linear relationship between 0.93 and 7.73 ng/mL. The LOD of the chemiluminescent ELISA was 0.12 ng/mL, and there was a good linear relationship between 0.29 and 2.68 ng/mL [14]. Although ELISA has the advantages of high sensitivity and exceptional convenience, it also has some disadvantages, such as relatively poor stability and high antibody costs. Molecularly imprinted polymers (MIPs) have been used in various analyses and sensors, and they have demonstrated great potential in the replacement of antibodies used in ELISA [15,16]. Munawar et al. proposed a competitive method based on molecularly imprinted polymer nanoparticles (MINAs), in which FB_1_ is fixed on a solid carrier for the preparation of nano MIPs, and the LOD was 1.9 pM in the concentration range of 10 pM–10 nM [5]. Qian et al. used a Max column combined with high-performance liquid chromatography–tandem mass spectrometry to detect FB_1_ in dairy products. Upon the addition of FB_1_ in a concentration range of 0.1–5 μg/L to milk, the authors revealed that the recovery was 76.4–92.3%.The LODs of FB_1_, FB_2_, and their hydrolytic metabolites were 0.03, 0.03, and 0.1 μg/L, respectively [17]. However, the above technologies involve high costs, and require significant human and material resources, which need to be analyzed by experienced researchers. The electrochemical method has relatively high sensitivity, low LOD, and strong specificity in target recognition. Zheng et al. established an electrochemical detection sensor. The sensor showed a good linear relationship at an FB_1_ concentration range of 1 × 10^−11^–1 × 10^−4^ g/mL, and the LOD was 10 pg/mL [18]. Based on the antigen antibody reaction and color development characteristics of colloidal gold, the immunochromatography assay (ICA), with colloidal gold as the medium, has been widely used in the field of rapid detection. Ren et al. developed a colloidal gold test strip for FB_1_ detection by ICA with an LOD of 2.5 ng/mL and used it for the rapid detection of FB_1_ in corn [19]. Because antibodies are expensive and other materials are in short supply, the research and development of test strips requires significant human and other resources. The above-mentioned methods have several advantages, namely low LOD and reliability; however, the cost is high, and on-site rapid detection is challenging. Therefore, it is necessary to introduce low-cost raw materials with strong affinity, and specificity, and an aptamer is a good choice.

Aptamers are single-stranded DNA or RNA that specifically bind to a target (small molecule or protein) with high affinity and selectivity [20]. Aptamers have the ability to amplify the signal of the sensor, and signal amplification is a key component of ultrasensitive analysis for the detection of trace amounts of analytical targets, such as various biologically significant macro and small molecules, disease biomarkers, toxins, bacterial pathogens or viruses, and environmental contamination [21]. Aptamers have also played a significant role in advancing cancer treatment, with alpha-PCNA (proliferating cell nuclear antigen) aptamers as therapeutic aptamers and adriamycin as a chemotherapeutic drug instrument delivered to tumor cells for precision treatment [22]. Aptamer biosensing systems have significantly improved the quantitative detection of molecular biomarkers of various harmful diseases, such as ovarian cancer. Farzin et al. developed a silver-nanoparticle-modified, kainamine-oxime-modified, polyacrylonitrile-nanofiber-based build for the sensitive detection of CA125 with an LOD of 0.0042 U/mL [23]. In the field of food safety detection, Talari et al. developed an optical probe nanosensor based on multiwalled carbon nanotubes and reduced graphene quantum dots for the specific detection of the organophosphorus pesticide diazinon with a detection limit of 0.4 nM (0.14 μg/L) [24].

The specific FB_1_ aptamer F10 with high affinity was successfully selected by the systematic evolution of ligands by exponential enrichment (SELEX) [25]. In this study, a fluorescent sensor was designed for the specific detection of FB_1_. F10 was labeled with a fluorescent molecule (FAM), and the fluorescence of F10 could be quenched by GO. GO is a type of nanomaterial with higher quenching efficiency. Compared with other quenchers, it can maintain the stability of biomolecules on its surface and prevent enzyme digestion [26]. As shown in Figure 1, in the absence of FB_1_, π-π stacking promoted FB_1_-specific aptamer F10 adsorption on the surface of GO. The fluorescence resonance energy transfer (FRET) interaction between fluorescent molecules and GO resulted in the quenching of the FAM fluorescent signal. The method had a good linear relationship in the FB_1_ concentration range of 0–3000 ng/mL, and the LOD was 14.42 ng/mL. The method was used to detect FB_1_ in corn flour samples, and recovery was high, consistent with the ELISA results.

## 2. Materials and Methods

### 2.1. Reagents and Apparatuses

FB_1_ was purchased from Binzhi Biotechnology (Shanghai, China). Ochratoxin A (OTA) was purchased from Xiheng Biotechnology (Shanghai, China). Zearalenone (ZEN) was purchased from Fubo Biotechnology (Beijing, China). Aflatoxin B_1_, aflatoxin G_1_, and aflatoxin G_2_ were purchased from Sigma-Aldrich Corporation (St. Louis, MO, USA). NaClK, Cl, and Na_2_HPO_4_·12H_2_O were purchased from Tianjin No. 3 Chemical Reagent Factory (Tianjin, China). CH_3_OH was purchased from Corus Chemical Co., Ltd. (Shanghai, China). KH_2_PO_4_ was purchased from Shanghai Aladdin Biochemical Technology Co., Ltd. (Shanghai, China). GO dispersion was purchased from Nanjing Xianfeng Nanomaterials Technology Co., Ltd. (Nanjing, China). All reagents used in this study were of analytical grade. Deionized water (18.2 MX) was used in buffer preparation, and 0.01 M phosphate-buffered saline (pH = 7.4, PBS) was used for fluorescence measurements. The corn flour samples were obtained from local markets. The aptamer used in this study was synthesized by Tsingke Biological Technology (Beijing, China). The sequence of the F10 was 5′-FAM-CGA TCT GGA TAT TAT TTT TGA TAC CCC TTT GGG GAG ACA T-3′. An Agilent G9800A spectrofluorometer (Agilent Technologies, Santa Clara, CA, USA) was used for obtaining the spectra. A Thermo BIOMATE 3S microplate reader (USA) was used for reading the absorbance of the ELISA.

### 2.2. Specificity of the Fluorescent Aptasensor for FB_1_ Detection

In order to verify the specificity of the fluorescence sensor, under the above optimized conditions, several other common mycotoxins that coexist with FB_1_ in corn grains (AFB_1_, AFG_1_, AFG_2_, ZEN, and OTA) were used as negative controls, and PBS was used as the blank control for the specificity study. FAM-F10 was added to the tubes to give a final concentration of 50 nM in a 3 mL system. AFB_1_, AFG_1_, AFG_2_, ZEN, OTA, FB_1_, and a toxin mixture were added to the tubes to give a final concentration of 200 ng/mL for each toxin. The final volume of each tube was adjusted to 3 mL with PBS (pH = 7.4). The mixed system was incubated in a constant-temperature shaker at 37 °C and 60 rpm for 10 min. GO, at a concentration of 46 μg/mL, was added to the system after 10 min of incubation, and the system was thoroughly mixed. The excitation wavelength was set to 497 nm, and the fluorescence value was measured at 520 nm.

### 2.3. Sensitivity of the Fluorescent Chemosensor for FB_1_ Detection

To further investigate the sensitivity of the fluorescence sensor and the linear relationship between toxin concentration and fluorescence values, different concentrations of FB_1_ were added to the system containing 50 nM FAM-F10. A concentration gradient of FB_1_ was established as follows: 0, 10, 15, 40, 80, 100, and 200 ng/mL. The final volume was made up to 3 mL by fixing with PBS. The mixed system was incubated in a constant-temperature shaker at 37 °C and 60 rpm for 10 min. GO, at a concentration of 46 μg/mL, was added to the system, and the system was thoroughly mixed. The excitation wavelength was set to 497 nm, and the fluorescence value was measured at 520 nm.

### 2.4. Analysis of FB_1_ in the Corn Flour Samples

Firstly, 10 g of the two corn powders was dissolved in 50 mL of methanol, and the solutions were thoroughly mixed and then centrifuged at 4000 rpm for 20 min at 4 °C. Next, the supernatant was collected and filtered through a 0.22 μm filter membrane and diluted three times with double-distilled water, 10 times each. The two samples, designated as 1 and 2, were stored at 4 °C. Standard FB_1_ solution was added into each tube, so that the final concentrations were 0, 20, 80, and 200 ng/mL, and PBS was used to adjust the volume to 3 mL. A final concentration of 50 nM of the aptamer FAM-F10 was added to each tube. The mixed system was incubated in a constant-temperature shaker at 37 °C and 60 rpm for 10 min. GO was added to the system at a concentration of 46 μg/mL, and the system was mixed to measure the fluorescence value. The parameters of the fluorometer were set to an excitation wavelength of 497 nm, and the fluorescence was recorded at 520 nm. An FB_1_ ELISA kit was used to validate the results of the spiked recovery experiments for both samples. To analyze the data, the above experiments were repeated three times independently.

## 3. Results and Discussion

### 3.1. Principle of Fluorescent Sensors

Modification of the FAM fluorescent tag at one end of the aptamer makes it highly fluorescent, and GO is an excellent fluorescent bursting agent. When the target FB_1_ does not exist in the system, the addition of FAM-F10 coupling can adsorb on the GO surface by p-p stacking, and the FRET between the fluorescent molecule and GO rapidly bursts the fluorescence and measures a low fluorescence value. When the target FB_1_ exists in the system, the FAM-F10 coupling binds to the target, thus greatly reducing the binding efficiency of GO and FAM and measuring a high fluorescence value. As shown in Figure 1A, when FAM-F10, GO, and FB_1_ were simultaneously added to the system, the fluorescence significantly increased. Therefore, it was demonstrated that the specific binding between FAM-F10 and FB_1_ could significantly reduce the binding efficiency of GO and FAM and the quenching efficiency of GO, thereby rendering the fluorescence unable to be completely quenched.

### 3.2. Optimization of the FAM-F10 Conjugate Concentration

Excessive FAM–aptamer conjugation can cause the fluorescence value to exceed the range that the fluorometer can detect. In order to make the measured fluorescence value within the detectable range, different concentrations of FAM-F10 were used for fluorescence detection. The aptamer FAM-F10 was added to the tubes to prepare final concentrations of 16.67, 33.33, 50.00, and 66.67 nM, and the volume was adjusted to 3 mL with PBS (pH 7.4). The excitation wavelength of the fluorescence spectrometer was set to 497 nm, and the fluorescence value was recorded at 520 nm. GO was added to the system to completely quench the fluorescence. As shown in Figure 1B, when the concentration of FAM-F10 was 50 nM, the fluorescence measurement at 520 nm was at the highest value that was within the detection range of the instrument. Therefore, 50 nM was selected as the optimal concentration of FAM-F10.

### 3.3. Optimization of GO Concentration

GO can burst the fluorescence of FAM fluorescent groups. In this study, as GO is able to completely burst the fluorescence of FAM, the presence of the target FB_1_ in the system could return the fluorescence value to achieve the detection of the minimum concentration of FB_1_ in the system, so the GO concentration was optimized. Under the optimal concentration of FAM-F10, GO was added to the tubes to prepare final concentrations of 40, 42, 44, 46, 48, and 50 μg/mL, and the volume was adjusted to 3 mL with PBS (pH = 7.4). The excitation wavelength of the fluorescence spectrometer was set to 497 nm, and the fluorescence value was recorded at 520 nm. High GO concentrations led to complete fluorescence quenching, while low GO concentrations led to incomplete quenching, resulting in the high fluorescence of the system. As shown in Figure 1C, GO, at a concentration of 46 μg/mL, was sufficient to completely quench the fluorescence. Therefore, this concentration was used as the optimal concentration of GO for subsequent detection assays.

### 3.4. Optimization of the Incubation Time

To make sure that FAM-F10 and FB_1_ were completely combined and the fluorescence value was stabilized, it was necessary to optimize the incubation time. Under the optimal concentration of FAM-F10, FB_1_ was added into the tubes to prepare a final concentration of 300 ng/mL, and then mixed for 1 min, followed by incubation in a constant-temperature shaking table at 37 °C and 60 rpm. The incubation time was set to 10, 20, 30, 40, 50, and 60 min. The excitation wavelength of the fluorescence spectrometer was set to 497 nm, and the fluorescence value was recorded at 520 nm. When the fluorescence values were stable, the target was completely bound to the aptamer. The optimization results are shown in Figure 1D. The fluorescence intensity gradually increased within 10 min and then stabilized. Therefore, the incubation time of FAM-F10 and FB_1_ was set to 10 min.

### 3.5. Specificity of the Fluorescent Aptasensor for FB_1_ Detection

Although the aptamer could specifically bind to the target, the established fluorescent sensor still required specific verification. Other mycotoxins coexisting with FB_1_ were used as controls to verify the selectivity of the fluorescent sensor. In this study, AFB_1_, AFG_1_, AFG_2_, ZEN, OTA, and all toxin mixtures, including FB_1_, were used as controls. The results are shown in Figure 2. The fluorescence measurements at 520 nm in the blank control group and the negative control groups of AFB_1_, AFG_1_, AFG_2_, ZEN, and OTA were at their lowest levels. The quenching efficiency of GO to FAM was the strongest, and the quenching efficiency was not reduced after the addition of toxins. In contrast, the fluorescence measurements at 520 nm in the toxin mixture and FB_1_ standard solution groups were significantly different from those of the other groups; the fluorescence values were significantly higher than those of the other groups. These results demonstrated that the sensor had strong specificity and selectivity.

### 3.6. Sensitivity of the Fluorescent Aptasensor for FB_1_ Detection

According to the principles of sensor detection, there should be a positive correlation between the concentration of FB_1_ in the system and the fluorescence value measured at 520 nm. With the gradual increase in the FB_1_ concentration in the system, the quenching efficiency of GO to FAM gradually decreased, and the fluorescence value measured at 520 nm accordingly increased. As shown in Figure 3A, by setting the gradient of FB_1_ and analyzing the fluorescence values of FB_1_ at different concentrations, a standard curve with the FB_1_ concentration as the abscissa and the fluorescence intensity as the ordinate could be obtained, in which R^2^ = 0.9936 and the equation of the standard curve was y = 0.2576x + 10.98. Figure 3B shows that when the concentration of FB_1_ was in the range of 0–3000 ng/mL, the fluorescence intensity had an obvious linear relationship with the FB_1_ concentration. The LOD and LOQ were calculated using the following data. The formula for the LOD was 3.3 × σ/S, where σ is the standard deviation of the test value of the blank group, and S is the slope of the standard curve. Similarly, the formula for the LOQ was 10 × σ/S. Under optimal conditions, the LOD of the method was 14.42 ng/mL, and the LOQ was 43.70 ng/mL.

A fluorescent sensor based on fluorescently labeled aptamer FAM-F10 and GO for FB_1_ detection was established. There are many methods to detect FB_1_, such as antibody-based flowmetric immunoassays with detection limits of 0.025 and 0.097 ng/mL in corn and feed, respectively [27]. In contrast with the reported antibody-based approach, a FAM fluorescent group was modified at the 5′ end of the FB_1_-specific binding aptamer F10 to provide fluorescence properties, and a GO with fluorescence bursting properties was subtly introduced. The aptamer used in this experiment has the significant advantages of low cost and good batch-to-batch consistency compared with the antibody. Compared with other methods using fluorescence burst [28], one of the advantages of this method is that it combines the fluorescent-group-labeled FAM with the aptamer, which ensures the stability of fluorescence and simplifies the experimental process by eliminating the step of incubating the aptamer with other materials [29]. Compared with LC-MC, the sample pre-treatment in this study is simpler and easier for actual sample detection [30]. Compared with the ELNOA method [31], the FRET method requires less time and fewer steps to measure FB_1_, which greatly saves time and costs. In terms of LOD, the detection limit of ELNOA is lower than that of FRET, but both methods can meet the detection requirements of the EU and FAD. The FRET method enables rapid on-site detection (POCT) due to the popularity of portable fluorometers.

### 3.7. Detection of FB_1_ in Corn Flour Ssamples

To verify the detection accuracy of the fluorescence sensor using actual samples, two types of corn flour were pretreated, and the FB_1_ standard solution was added to the two samples to prepare FB_1_ concentrations of 0, 20, 80, and 200 ng/mL. The fluorescence sensor and ELISA generated in this study were used for standard addition and recovery experiments. As shown in Table 1, the recovery of corn flour sample 1 measured by this method was 89.13–98.66%, and the recovery detected by the ELISA was 98.35–100%. The recovery of corn flour sample 2 was 90.7–102.08%, and the recovery by the ELISA was 97.73–99.42%. Thus, it was confirmed that the results obtained by the constructed sensor were highly consistent with the detection results of the ELISA, which proved the accuracy of the detection results. Therefore, it was verified that the proposed sensor can be used to detect the content of FB_1_ in actual samples.

## 4. Conclusions

In this study, the FRET method was established by fluorescently labeling FAM on an FB_1_-specific nucleic acid aptamer and using the ability of GO to burst the fluorescence to establish a fluorescent sensor. By optimizing the GO concentration, under the premise that GO is able to burst the fluorescence of FAM, as long as a trace amount of FB_1_ is present in the system, FB_1_ can enter into specific binding with F10, so that GO cannot burst the fluorescence of FAM, and its fluorescence value is significantly rebounded compared with the fluorescence value at complete burst. The method is characterized by high sensitivity and wide detection range, with good linearity in the concentration range of 0–3000 ng/mL. LOD was as low as 14.42 ng/mL, and the spiked recovery experiments confirmed that the method could efficiently detect FB_1_ in maize samples with recoveries of 89.13–102.08%. Based on the excellent properties of the aptamer, the established fluorescent sensor provides a new idea for the design of other target detection methods with a wide range of applications.

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
