# Peer review of "Determination of Fumonisin B1 by Aptamer-Based Fluorescence Resonance Energy Transfer"

_sensors, 2022, doi:10.3390/s22228598_

Round 1

Reviewer 1 Report

1. More details in Method 2.2 should be added. It will benefit people who like to use this strategy.

2. Compare the cons and pros of the current strategy to your recent publication Zhao et al, Sensor 2022

3. Inconsistent of FB1 and FB1 word in the manuscript, please stick to one format

Author Response

Point 1: More details in Method 2.2 should be added. It will benefit people who like to use this strategy.

Response 1: Thank you for your suggestion, we have added the details in Materials and Methods 2.2. “In order to verify the specificity of the fluorescence sensor, under the above optimized conditions, several other common mycotoxins that coexist with FB1 in corn grains: AFB1, AFG1, AFG2, ZEN and OTA were used as negative controls, and PBS was used as blank control for the specificity study.”

Point 2: Compare the cons and pros of the current strategy to your recent publication Zhao et al, Sensor 2022

Response 2: FRET detection of FB1 requires less time and fewer steps, which greatly saves time costs. The FRET method enables rapid on-site detection (POCT) due to the popularity of portable fluorometers.

Point 3: Inconsistent of FB1 and FB1 word in the manuscript, please stick to one format

Response 3: Thank you for the reminder, we have standardized the format of FB1 to FB1.

Reviewer 2 Report

General comment:

In this manuscript, authors developed a FRET method for the sensing of FB1 using aptamer. The research work was straight forward and clearly showed the specific detection of FB1. A few issues on presentations were identified as mentioned below. A minor revision is suggested for authors to address these issues.

Specific comments:

1.      Page 2: What was PCNA? Please do not expect reader to understand all abbreviations

2.      Figure 3B: It is suggested that authors try to find out the binding relationship between FB1 and aptamer. Then use an equation to fit the data instead connecting the data points with meaningless straight lines.

3.      Line 267: “fluorescent sensor”, Line 269: “fluorescence sensor”. Please proofread carefully for accurate writing.

4.      Table 1. Please state the name of method in legend and also the headings in the table. Do not just use “this method”. It was a little confusing at the first glance.

5.      Table 1. How did the spiked value compare to the actual amount of FB1 that existed in natural corn sample?

Author Response

Point 1: Page 2: What was PCNA? Please do not expect reader to understand all abbreviations.

Response 1: Thank you for your suggestion, the full name of PCNA has been written in the manuscript.

Point 2: Figure 3B: It is suggested that authors try to find out the binding relationship between FB1 and aptamer. Then use an equation to fit the data instead connecting the data points with meaningless straight lines.

Response 2: Thank you very much for your suggestion. The aptamer described in this paper is the specific aptamer F10 of FB1 screened in the cited literature, and the affinity of the aptamer has been calculated by the relationship between FB1 and the aptamer in the cited article. In contrast, our study is based on the aptamer to establish a method to detect FB1. Figure 3B makes a fitted curve by different concentrations of the target corresponding to different fluorescence intensities, and the detection range of the method is derived from the curve.

Point 3: Line 267: “fluorescent sensor”, Line 269: “fluorescence sensor”. Please proofread carefully for accurate writing.

Response 3: Thank you for the reminder, We carefully calibrated and changed the fluorescent sensor to fluorescence sensor.

Point 4: Table 1. Please state the name of method in legend and also the headings in the table. Do not just use “this method”. It was a little confusing at the first glance.

Response 4: Thank you for your suggestion, we have modified it in the legend.

Point 5: Table 1. How did the spiked value compare to the actual amount of FB1 that existed in natural corn sample?

Response 5: Since the actual samples were tested with 1 g of corn sample dissolved in 5 mL of solution, the FB1 limits of 200-2000 μg/kg specified for corn flour could be converted. With 200μg/kg as the calibration value, 1kg of corn flour was dissolved in 5L solution, the upper limit of FB1 was converted to 40-400 ng/mL. The actual samples tested in this study were spiked in amounts that included both low and high concentrations to meet the detection requirements of the standard.

Round 2

Reviewer 1 Report

.